# Optimization of the Composition of Toluene-Based Liquid Scintillator

**Dmitriy Beznosko [1,*]** , **Elijah Holloway [1]** **and Alexander Iakovlev [2]**

1   Chemistry and Physics Department, Clayton State University, 2000 Clayton State Blvd., Morrow, GA 30260, USA
2   Community Christian School, 2001 Jodeco Rd., Stockbridge, GA 30281, USA
*   Correspondence: dmitriybeznosko@clayton.edu

**Abstract:** Scintillators in general and organic liquid scintillator specifically are widely used as a medium for the detection of charged particles for numerous applications in science, medicine, engineering, and other areas. The composition of the scintillator affects not only its direct performance characteristics, but also the overall cost. Optimization of this composition provides the ability to design particle detectors with an optimized light yield and emission spectra of the detection medium while optimizing the expenses at the same time. This article describes work on toluene-based liquid scintillator component optimization, where PPO is used as a fluor and POPOP as a shifter. The light yield vs. concentration and the changes in the output spectra will be presented. The empirical fit of the output spectrum using the measured contributions of the components is discussed. Further plans include the light attenuation measurements for different compositions.

**Keywords:** scintillator; light emission spectra; light emission time; composition optimization

## 1. Introduction

The history of scintillators is very long in science and dates to the first-most particle detectors. Nowadays, the advances in both the photon detectors that are used in the conjuncture with scintillators, and the increase in the particle flux for the most typical uses of detectors in accelerators and similar fields have placed requirements of the short light emission time and the high light yield from the scintillator. The matching of the output light spectrum is also important to match the detector most sensitive region. Even simple designs of modern particle detectors must account for these details [1]. Specifically important is the light emission time by the scintillator as this affects the detector performance in the conditions of the high particle flux, which can happen not only at the accelerator facilities but even for measurements of the ultra-high energy cosmic rays [2] that is a current research area actively looking for possible new physics appearance [3].

Liquid organic scintillator has both the short rise and decay times of the emitted light, and sufficiently high light yield that is suited for detection of charged particles. It is similar to plastic scintillator in properties but is more cost effective per unit mass and is harder to handle due to high flammability. Wavelength shifter additives can increase the overall light detection by a specific photodetector by changing the scintillator output spectrum into the best detection region. New materials, such as novel water-based liquid scintillator material, aim to reduce the cost and provide better control over the light yield while reducing flammability [4]. In addition, scintillator composition affects the output light pulse width that is important for any timing measurements using scintillator-based detectors.

The composition of a scintillator affects not only its performance, but also the overall cost that includes the price of all the components. Typically, besides the scintillator itself, two additional components are used. Both are the wavelength shifters since the organic scintillation light is initially in the high UV region (typically around 200 nm), and most photon detectors are most sensitive in the 'blue' region centered around 420 nm or so. The primary

shifter, commonly called a fluor, that is used in this work, is PPO (2,5-diphenyloxazole). A secondary shifter used is POPOP (1,4-di-(5-phenyl-2-oxazolyl)-benzene)—commonly just called a shifter. The base scintillator is toluene (methylbenzene).

The novelty of the work presented is the measurements of the PMT response time to the light output duration of the scintillator as well as the successful search for the optimal concentrations of the dopants. The shift in the output spectra is also highlighted and it is the subject of the current research. The toluene is used as the base scintillator. The previous results [5,6] concentrate on the spectral changes and use different scintillator base and a secondary shifter.

## 2. Setup and Methodology

The 'dark box' is used to house the experimental setup consisting of two Photo-Multiplier Tubes (PMT) and a 3D printed sample holder as shown in Figure 1. The PMT on the right in the Figure 1 is Hamamatsu R580 (Hamamatsu, Bridgewater, NJ, USA), and the PMT on the left is MELTZ FEU-115 (MELTZ, Moskva, Russia). The choice is based on the differences in the spectral sensitivity of these PMTs. The FEU-115 is an older model that is not sensitive to the UV region as shown in Figure 2a, whereas the R580 PMT has a range that is UV extended as seen in Figure 2b.

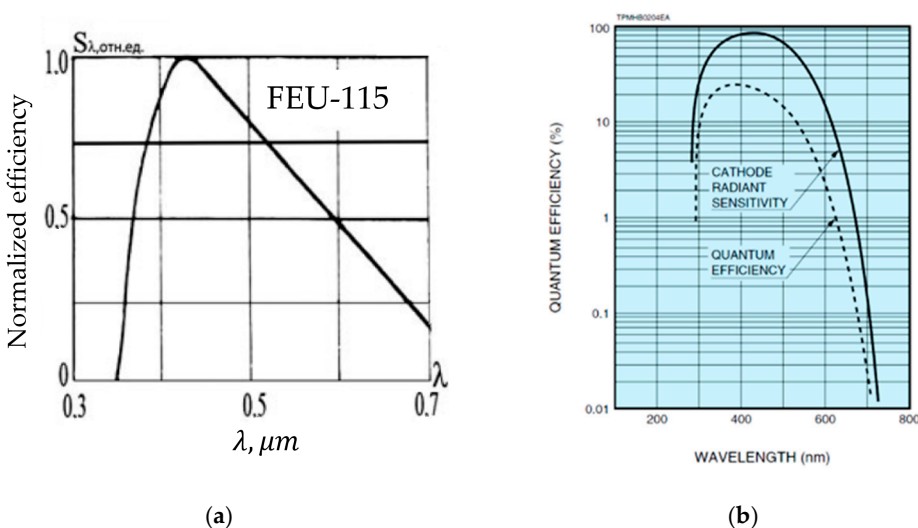

**Figure 1.** The experimental setup with two PMTs and a sample holder in a dark box.

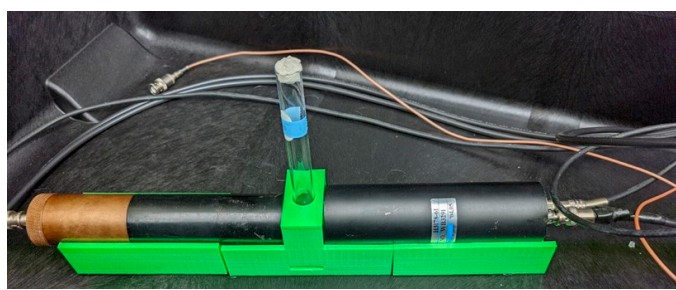

| (a) | (b) |
|:---:|:---:|

**Figure 2.** Detection efficiency vs. wavelength for: (**a**) FEU-115 PMT, (**b**) R580 PMT.

The data acquisition system consists of a CAEN DT5730 ADC (Analog-to-Digital Converter) (CAEN, Viareggio Lucca, Italy) that has eight channels and records at 500 mega-samples per second. The waveform recorded by each ADC channel consists minimum of 5110 points, each point every 2 ns, for a total of 10.22 µs. The internal memory of the device is used as an event buffer which can contain up to 1024 events for 10.22 µs setting. The full range of amplitude digitization for this ADC is $2^{14}$ bits with a signal amplitude of $\pm 1$ V, or

from 0 to −2 V. Data from the ADC is read via a USB2 port. The compressed binary file size for one event is ~50 KB. The PMT biases were chosen to be 1600 V for the R580 and 1750 V for FEU-115. The choice is based on avoiding the ADC saturation at the highest light output of the scintillator as well as on reducing the PMT noise. The components were ordered from Sigma-Aldrich (Sigma-Aldrich Inc., Burlington, MA, USA) as: Toluene ACS reagent ≥99.5%, PPO 99% suitable for scintillation, and POPOP 99% suitable for scintillation.

The back-to-back setup design [7] features a coincidence trigger between the two PMTs within a 50 ns window that reduced any external noises in the system and allows to obtain two data sets from different detectors. The small sample size and the quick data taking process with the ADC over only a few seconds reduce a contribution from cosmic rays to negligible amount (the cosmic rays' flux is ~1 muon per $cm^2$ per minute per steradian). To excite the scintillator, $^{90}$Sr source is used. The full setup is shown in Figure 3.

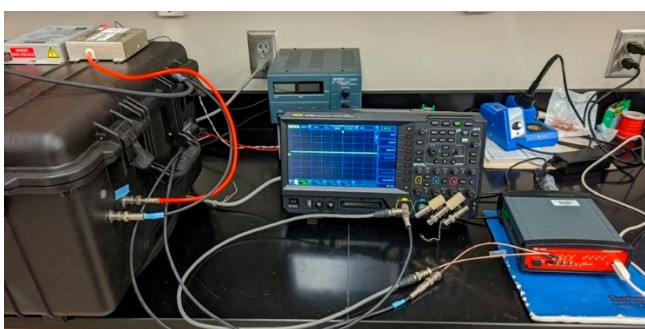

**Figure 3.** Full experimental setup showing the dark box and the ADC.

To measure the absorption and emission spectra of the scintillator, the following devices were used: Agilent Cary 5000 UV-Vis-NIR Spectrophotometer (Agilent, Santa Clara, CA, USA) to study the absorption characteristics of the different scintillator components at different wavelengths, and the PerkinElmer LS 55 Luminescence Spectrometer (PerkinElmer, Waltham, MA, USA) to measure the output spectra of different scintillator samples when excited by the UV light (~200 nm).

### 3. Results

The experimental results will be presented as three groups: the PPO concentration results, the POPOP concentration results and the preliminary results for the output spectrum measurements.

### 3.1. PPO Results

The PMT response with different fluor concentrations was measured to optimize the concentrations of PPO. The total light output measured by the R580 PMT is presented in Figure 4. The result from the FEU-115 PMT is not shown as the shape of the total response for this PMT is the same as for the R580 PMT with the matching peak at 7 g/L. Here, the PMT response is the maximum pulse height in the ADC bins units.

Additionally, the measurements of the PMT pulse width change with the PPO concentration were conducted. The pulse width is defined here as the time evaluated at 50% (half of the width) and 80% (full width) of the total area under the pulse, since a median is a statistically solid measure that is also applicable if the pulse shape is distorted by the long transmission line [2,8]. The results for R580 PMT only are shown because it has the smallest signal spread during the electron transit and thus is most sensitive to the pulse width change. The measurement results are shown in Figure 5. Error bars are ~0.1 ns and are not clearly visible on the plot. The high values of pulse width at very low PPO concentrations most likely are due to low signal.

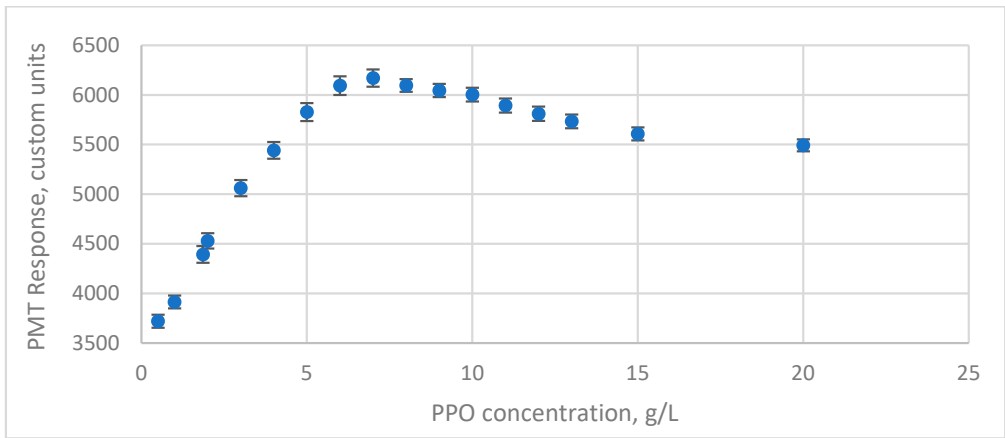

**Figure 4.** R580 PMT response to scintillator light for different PPO concentrations.

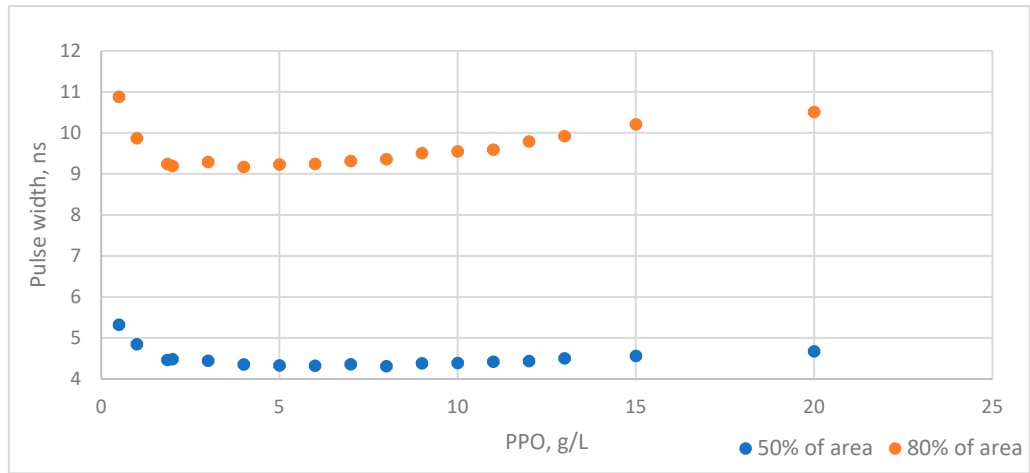

**Figure 5.** R580 PMT pulse width (50% and 80% of the total area) for different PPO concentrations. Error bars are ~0.1 ns and are not clearly visible.

### *3.2. POPOP Results*

The POPOP shifter has been added at different concentrations to 10 g/L PPO solution to assess the change in the output of both PMTs. Note that the addition of the POPOP shifter makes the overall response of the R580PMT and FEU115 differ. The pulse width change was insignificant within the measurement error (~0.1 ns) and is not shown.

The response of R580 PMT to the scintillator light at different POPOP concentration is shown in Figure 6. Note that the response is reducing with the concentration increase.

From Figure 7 we note that the response of FEU-115 is increasing with the POPOP concentration and has a clear peak value. This can be explained by the fact that R580 PMT is a UV sensitive detector, while FEU-115 is not. The addition of the shifter absorbs some of the light in the UV region and shifts it into the blue region, at certain efficiency. Thus, the overall amount of light is lost, but the increased sensitivity for FEU-115 compensates in part for this loss.

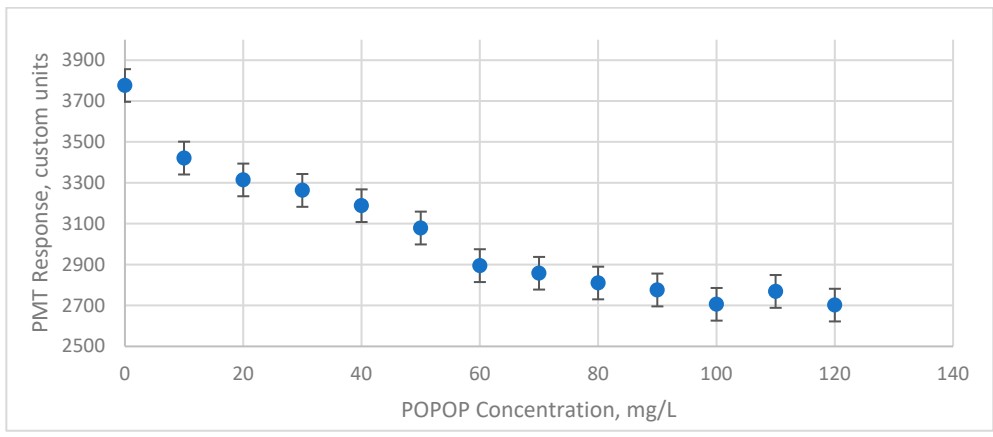

**Figure 6.** R580 PMT response for different POPOP concentrations in a 10 g/L PPO scintillator.

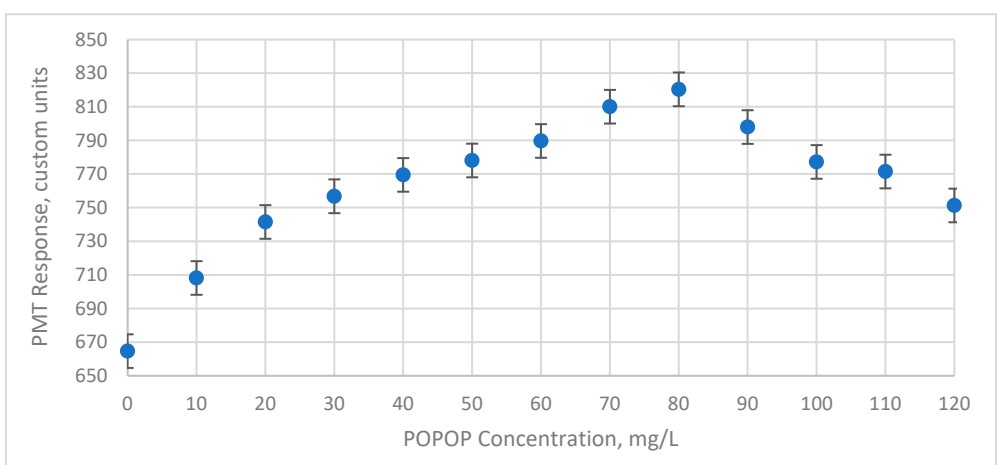

**Figure 7.** FEU-115 PMT response for different POPOP concentrations in a 10 g/L PPO scintillator.

*3.3. Output Spectrum Measurements*

To verify the explanation of results for the POPOP measurements with the two PMTs, the LS 55 Luminescence Spectrometer was used to measure the scintillator light output spectrum for different POPOP concentrations when excited by ~200 nm light. This light would excite primarily the PPO due to its much higher concentration. PPO will emit the same light as when excited by the charged particle passage. Then POPOP will absorb this light that is emitted by the PPO and the resulting total output will have the same spectrum that the PMT is observing. Some UV light is also directly absorbed by the POPOP, so the total amount of light is not conserved in this measurement.

This change in the output spectrum of the scintillator vs. the POPOP concentration is presented in Figure 8. Starting with the reference spectrum of pure PPO at 10 g/L concentration, POPOP is added, and spectra are recorded. For lower concentrations of 0.03 g/L to 0.05 g/L, the change in the shape is visible. After certain value of about 0.1 g/L and above, the overall change in the output spectrum is not significant as all light from the PPO is already converted and only the total amplitude is increasing due to the higher absorption of the 200 nm light that is used for the excitation. Note that the 200 nm wavelength was chosen from the absorption properties of both the PPO and POPOP that were measured using the Cary 5000 UV-Vis-NIR Spectrophotometer.

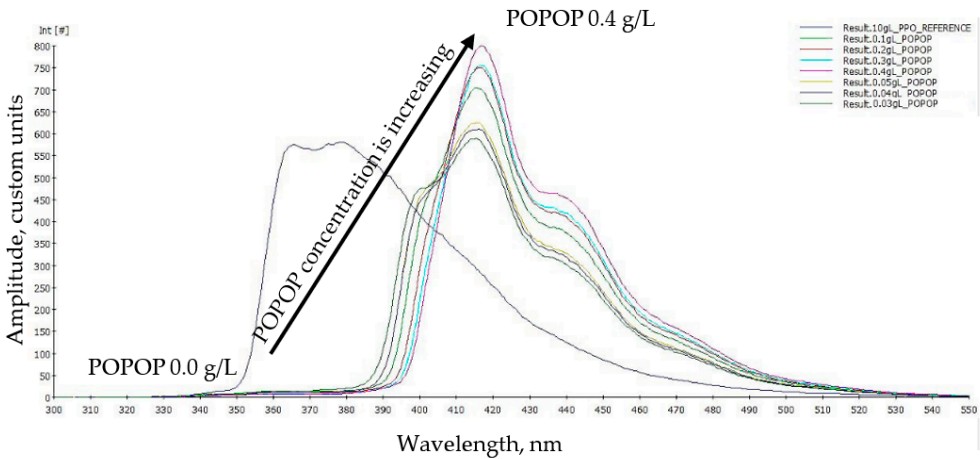

**Figure 8.** Scintillator output light spectrum for different POPOP concentrations.

## 4. Discussion

The overall trend is the overall shift of the light that is emitted by the PPO in the 350–390 nm range to the 400–460 nm range by the POPOP shifter. In the absence of the excitation light, the total integral under the curse (i.e., the total number of photons emitted) should reduce as POPOP doesn't have perfect efficiency. Thus, the UV sensitive PMT will see an overall light reduction as observed in Figure 6. The blue sensitive PMT would first benefit from this wavelength shift, as evident from Figure 7, but eventually the total light loss will be evident as well. Therefore, the explanation for the different in detection behavior of R580 and FEU-115 PMTs is well supported by Figure 8.

The commercial scintillators that use PPO and POPOP as dopants normally come with a fixed composition that can't be adjusted on request. This work should be helpful in choosing the certain ready formulations suited for the experiment purpose as well.

## 5. Conclusions

The measurements of the light output spectra from the liquid scintillator that is using PPO fluor and POPOP shifter as dopants produces several findings.

The concentration of the PPO fluor has a maximum light output, after that the light intensity decreases, most likely to light self-absorption by PPO itself.

Around the concentration range of the highest light yield, the time of light emission by the scintillator is weakly affected and grows only with high concentrations.

With some UV sensitive PMTs, the addition of the (secondary) shifter such as POPOP is not necessary. This makes the total scintillator production simpler and reduces costs.

With other PMTs, the addition of the POPOP is beneficial for the total signal detection up to a certain concentration.

The future plans include the measurements of the light attenuation length in the liquid scintillator vs. the dopants' concentration.

**Author Contributions:** Conceptualization, D.B.; methodology, D.B.; software, E.H.; validation, A.I.; formal analysis, D.B. and E.H.; investigation, D.B.; resources, A.I.; writing—original draft preparation, D.B.; writing—review and editing, A.I. All authors have read and agreed to the published version of the manuscript.

**Funding:** This research was funded by Clayton State University, College of Arts & Sciences Minigrant, FY 2021–22.

**Data Availability Statement:** Data is available on request due to restrictions as the experimental work and analysis are not yet completed in full.

**Conflicts of Interest:** The authors declare no conflict of interest.

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
