# Peer review of "Optimization of the Composition of Toluene-Based Liquid Scintillator"

_instruments, doi:10.3390/instruments6040056_

Round 1
Reviewer 1 Report
This article describes the optimization of fluor -PPO- and shifter -POPOP- concentration in the toluene-based scintillator. Liquid scintillators containing PPO and POPOP have been widely studied for a long time, and the novelty of the present study is not clearly described. I would reconsider for the publication after major revision. The following comments should be addressed:
1. Please provide sufficient background and include all relevant references in the section of the introduction. In particular, previous research trends on PPO and POPOP-containing liquid scintillators should be presented.
2. The description for the methods is not sufficient. Please include information about the materials used in this work, such as vendor, purity, etc. please explain how you calculate the light output of the scintillators. If a pulse height or pulse integral spectrum is obtained, it should be displayed. Add the information about the bias voltage of the PMT used.
3. The scintillation light output of the scintillator developed in this study is preferably compared to the light output of commercially available liquid scintillators.
Author Response
Dear reviewer
thank you for your comments. please find the detailed responses below:
- Please provide sufficient background and include all relevant references in the section of the introduction. In particular, previous research trends on PPO and POPOP-containing liquid scintillators should be presented.
---added the last paragraph to the introduction and two additional refereences that highlight the previous work on pulse duration and spectral output :The novelty of the work presented is the measurements of the PMT response time to the light output duration of the scintillator as well as the successful search for the optimal concentrations of the dopants. The shift in the output spectra is also highlighted and it is the subject of the current research. The toluene is used as the base scintillator. The previous results [5, 6] concentrate on the spectral changes and use different scintillator base and a secondary shifter.
-Please note that while there is a lot of work about scintillator done, most of that is not published as a separate article and is a part of detector design papers in NIM and similar as scintillator work is typically done for a specific detector. The goal of this article is to help future experiments to make pre-choices before starting their testing thus saving time and funding.
2. The description for the methods is not sufficient. Please include information about the materials used in this work, such as vendor, purity, etc. please explain how you calculate the light output of the scintillators. If a pulse height or pulse integral spectrum is obtained, it should be displayed. Add the information about the bias voltage of the PMT used.
--added the following passages:
--The PMT biases were chosen to be 1600V for the R580 and 1750V for FEU-115. The choice is based on avoiding the ADC saturation at the highest light output of the scintillator as well as on reducing the PMT noise.
--The components were ordered from Sigma-Aldrich [10] as: Toluene ACS reagent ≥99.5%, PPO 99% suitable for scintillation, and POPOP 99% suitable for scintillation.
--Here, the PMT response is the maximum pulse height in the ADC bins units.
- The scintillation light output of the scintillator developed in this study is preferably compared to the light output of commercially available liquid scintillators.
added:
The commercial scintillators that use PPO and POPOP as dopants normally come with a fixed composition that can’t be adjusted on request. This work should be helpful in choosing the certain ready formulations suited for the experiment purpose as well.
-- commercial scintillator is produced with a variety of dopants and bases, thus direct comparison is not possible. Even those with PPO and POPOP have different % of each. If you compare the claimed by manufacturer values to the measured in terms of pulse duration they do match. but the light output we measure in custom units and manufacturers normally give an absolute value of in relation to antracit s the comparison in our opinion is not very useful. rather our work can be used to select certain products. for example - why choose PPO >10 g/L concentration (or equivalent for plastic) if its after the maximum light yield? normally, manufacturers do their optimization but they don't provide details as to how this was achieved. Which has prompted the current research.
Reviewer 2 Report
The article "Optimization of the composition of toluene-based liquid scintillator" is clear and well written and deserves to be published in Istruments Journal.
There are just some minor typos:
Line 69 there is a “.” after noises
Line 72 cm2
The sentence at Line 99 “Error bars are ~0.1 ns and are not clearly visible on the plot” has to added also in Figure 5.
Author Response
Dear reviewer:
thank you for your comments and the overall statement about our publication. below are the responses to your comments:
There are just some minor typos:
Line 69 there is a “.” after noises - removed.
Line 72 cm2 - fixed to uppercase
The sentence at Line 99 “Error bars are ~0.1 ns and are not clearly visible on the plot” has to added also in Figure 5. - added to caption.
Round 2
Reviewer 1 Report
The description of the introduction and methods is now adequate (hopefully more references are required...).
The manuscript will be accepted for publication after minor revision.
- Fig. 5, 6: Y-label "ADC Responce" -> "PMT Response" ?
- 5. A Batyrkhanov et al. -> 5. D Beznosko et al.,
Author Response
Dear reviewer
thank you for your helpful comments and suggestions. Below is the list of the fixes following your comments:
-----
The description of the introduction and methods is now adequate (hopefully more references are required...).
--thank you. we feel that adding more references will overload this section. Some references will appear later in text, like the chemicals manufacturer.
The manuscript will be accepted for publication after minor revision.
- Fig. 5, 6: Y-label "ADC Responce" -> "PMT Response" ?
---yes, thank you for noting. label is fixed for both figures!
- 5. A Batyrkhanov et al. -> 5. D Beznosko et al.,
---fixed